# Effect of Nitrogen Fertilization on Tree Growth and Nutrient Content in Soil and Cherry Leaves (*Prunus cerasus* L.)

Krzysztof Rutkowski [ID] and Grzegorz P. Łysiak *[ID]

Department of Ornamental Plants, Dendrology and Pomology, Poznan University of Life Sciences, Dąbrowskiego 159, 60-594 Poznań, Poland
* Correspondence: glysiak@up.poznan.pl; Tel.: +48-61-848-7946

**Abstract:** Nitrogen fertilization ensures the proper growth of trees. The aim of the study was to evaluate the impact of differentiated nitrogen fertilization on selected parameters. It was assumed that such analysis is an indirect picture of the needs of cherries grown in herbicide fallow. The content of minerals in two layers of the soil, in leaves, and its influence on tree growth, and the content of chlorophyll in leaves were assessed. The experiments were carried out in three different cherry orchards. Three levels of fertilization were applied in each orchard: 0 kg, 60 kg, and 120 kg N ha$^{-1}$. As expected the fertilization resulted in an increase in the content of nitrate and ammonium forms of nitrogen in the soil, however, their content was also dependent on precipitation and temperature. Additionally, high nitrogen fertilization increased the content of phosphorus and potassium and decreased the magnesium in the topsoil layer. High nitrogen fertilization caused the decreased content of phosphorus and potassium in the leaves. The level of calcium and magnesium in leaves increased with fertilization of 60 kg N ha$^{-1}$ but decreased with the dose to 120 kg N ha$^{-1}$. The use of nitrogen fertilization increased the vegetative growth of trees measured by leaf area and trunk cross-sectional area. However, the chlorophyll content was not dependent on the amount of nitrogen fertilization. Based on the results, it can be concluded that 60 kg N ha$^{-1}$ is the optimal dose, ensuring proper nutrition of cherry trees.

**Keywords:** *Prunus cerasus*; ammonium nitrate; chlorophyll; tree cross sectional area; mineral content; carotenoids





## 1. Introduction

The extensive root system allows plants to take up almost all nutrients from the soil, which makes soil fertilization the basic way of their nutrition. The availability of minerals depends on many factors. The main ones are the soil formation, the content of organic matter, the pH, and the course of climatic conditions. These factors, by inducing microbiological and chemical processes in the soil, affect their uptake [1].

Fruit plants are perennial plants meaning that they are characterized by lower nutritional requirements than annual plants. It is caused by a much larger and more durable root system, as well as the ability to store nutrients in permanent organs such as roots, trunks, branches, shoots, and buds. Fruit plants can meet nutritional needs even with a low concentration of nutritional compounds. The annual increase in stocks is difficult to estimate because it is impossible to calculate how many components collected were used to build new tissues and how much was 'in stock' [2,3]. Some of the collected components return to the soil with fallen leaves and cut shoots, and other parts are removed during agrotechnical work, such as flowers or fruit buds [4]. Orchard species differ in the demand for nitrogen. It depends mainly on the nitrogen content in the fruit and the weight of the entire crop relative to the weight of the rest of the plant. The least nitrogen contains apples, about 0.9 g for each kg, and the most kiwi because as much as 4.5 g per kg of fruit in a previous study [3]. Assuming that the average kiwi yield was approximately 20 t ha$^{-1}$ [5],

to compensate for the loss of nitrogen, resulting from the yield, fertilization should be 90 kg ha$^{-1}$. On the other hand, apples, although they contained much less nitrogen, the yield of apple trees was higher. In Poland, on many farms, the yield was 50 t ha$^{-1}$ in some, it might exceed 80 t ha$^{-1}$ [6,7]. Along with the crop, from 50 to 80 kg of N ha$^{-1}$ are taken out from the orchard. Therefore, the amount of nitrogen fertilization should depend on the yield of trees. According to the integrated cherry cultivation methodology, if spring frost is expected lower yield, for instance, the dose of nitrogen fertilization should be reduced by 30–50% [8].

When cultivating fruit plants, it is very important to maintain a balance between vegetative growth and the fruiting of trees [9], which allows for regular and annual fruiting. Excessive nitrogen fertilization increases growth and causes a deterioration in the quality of fruits and their storability [10]. Nitrogen, on the other hand, increases the weight of the fruit. The market requires the supply of large fruits with very good quality parameters. However, in the case of pome fruits, increasing nitrogen fertilization contributes to the deterioration of color and greater sensitivity to storage diseases. Stone species react weaker than pome species to high nitrogen fertilization [10]. This is due to an imbalance of minerals in the plant, which interferes with the growth and yield of trees. Therefore, the amount of nitrogen fertilization should depend on the expected yield in a given year [3], which varies with the weather conditions [11,12]. It is recommended to divide the planned dose of nitrogen fertilization into two or three parts. Splitting of nitrogen fertilization allows you to modify fertilization if factors that limit the fruiting, e.g., frost or weather, are unfavorable for pollination during flowering [3].

High nitrogen fertilization also increases susceptibility to field diseases by stimulating vegetative growth [3]. In cherry orchards, the recommended fertilizer dose depends on the organic matter. The higher the content of organic matter, the lower the dose should be applied, because the organic matter increases the efficiency of fertilizer use. According to the recommendations in cherry orchards fully fruiting, where the content of organic matter in the soil was 0.5–1.5%, the annual dose of nitrogen should be between 60 and 80 kg N ha$^{-1}$ [13,14]. If the content of organic matter in the soil is in the range of 1.6–2.5%, the dose may be reduced to 40–60 kg N ha$^{-1}$. When organic matter is in the range of 2.6–3.5%, the annual dose of nitrogen per hectare in a cherry orchard can be 20–40 kg [8]. With a yield per hectare of full fruiting (yield 15 t ha$^{-1}$), approx. 45 kg N [15] is removed from the orchard with the crop. The use of nitrogen applied in early spring by trees is small and usually does not exceed more than 10–12 percent. Such a small use means that during flowering there is still a lot of fertilizer nitrogen in the soil as research shows, in the apple orchard during the flowering period there was still in the soil nearly 60% nitrogen applied in spring, and there was still nearly 60% of applied nitrogen in the soil remaining in spring [16]. In the early stages of growth, plants use nitrogen taken up and accumulated in tissues in the previous season. The amount of nitrogen stored depends on the age of the tree size and applied fertilization [3]. In deciduous trees, before the end of the growing season and leaf fall, nitrogen is withdrawn from the leaves and transferred to the trunk, shoots, and older roots. The tree uses the stored nitrogen at the beginning of vegetation before it begins to take up from the soil [17,18]. Furthermore, fallen leaves decompose and the tree can reuse nitrogen after undergoing decomposition and mineralization processes mineralization [19], in the second year after the drop fall [20].

The source of nitrogen is organic matter decomposition and mineralization caused by the activity of heterotrophic aerobic bacteria, including, among others, bacteria from the genera *Azotobacter* and *Azospirillum*. Bacteria assimilate N$_2$ only for the purpose of cell metabolism, so they do not secrete bound N into the environment. Nitrogen enters the soil environment when the bacterial cells die. The amount of nitrogen that is supplied by microorganisms to the soil is estimated at several kilograms of N·ha$^{-1}$ per year [21]. Despite the small amount, the quantities supplied are of great importance for soil fertility. However, their occurrence and abundance depend on various environmental factors such as organic matter content, moisture, soil reaction, and climatic condition [22]. For example,

*Azotobater* is very high in soil above pH 6 and rarely occurs in soils with a pH below 6 [23]. In contrast, *Azospirilllum* naturally occurs in regions with warm and or hot climates [24].

Nitrogen fertilization leads to an increase in the size of the tree and the area of the leaves [25] and on the other hand, it limits the intensity of flowering [26]. Trees grown under low nitrogen availability have a lower rate of photosynthesis, which results in a lower yield and fruit size [27] because photosynthesis is a function of the surface of the leaves, not their mass [28]. A significant correlation was found between the specific leaf weight (SLA-specific leaf area) (leaf weight/area) and the nitrogen content of the leaves [29].

Nitrogen should be available throughout the growing season, so the sorption of mineral nitrogen is important. Plant roots are capable of absorbing nitrogen in the nitrate form ($NO_3^-$), as well as in the ammonium form ($NH_4^+$), but in well-aerated soils, nitrates are the dominant source of nitrogen [30]. Ammonium nitrogen ($NH_4^+$) as a cation is well absorbed interchangeably, so it does not leach into deeper layers of the soil. In contrast, nitrate nitrogen ($NO_3^-$) as an anion is not subject to biological and exchange sorption. It is washed out into the deeper soil layers beyond the reach of the root system [31]. This causes its loss as a nutrient and is also a source of groundwater contamination. Additionally, in the oxidation process of ammonium ions and their reduction to nitrite, nitrous oxide ($N_2O$) is formed, which is one of the factors that cause the degradation of the ozone layer in the atmosphere [32,33]. Therefore, optimal nitrogen fertilization must be adjusted to ensure a high yield with the lowest possible fertilizer dose [34]. One of the recommended solutions to reduce nitrogen loss is to divide the full dose of nitrogen into 2–4 components [15].

The intensity of photosynthesis is directly proportional to the chlorophyll content in the leaves. Chlorophyll content depends on the availability of light, the vigor of tree growth, species, cultivar and rootstock, and stress factors [35,36]. The chlorophyll content of the leaves is thickened when poorly lit, the central part of their crown is on average 1.3 times thicker than in the leaves on the outskirts of the crown [37]. The mechanism of their adaptation explains the higher concentration of green pigments in poorly lit leaves to worse light conditions. A Lack of light causes a decrease in the thickness of the leaf blade [38]. However, the conversion of chlorophyll concentrations per m$^2$ LSA (leaf surface area) shows that leaves from the peripheral part of the crown have the same or higher chlorophyll content as the middle [37].

The purpose of the study was to evaluate the impact of different doses of nitrogen fertilization on the content of the basic element in the soil and in the leaves and its influence on selected cherry growth parameters.

## 2. Materials and Methods

### 2.1. Location and Main Agrotechnical Treatments

The research was carried out in the years 2010–2013 on sour cherries (*Prunus cerasus* L.) cultivar 'Łutówka' grafted on rootstock (*Prunus mahaleb* L.). The trees were planted in spring 1999 (OR1), 2001 (OR2), and 2002 (OR3) on the experimental farm of the Poznań University of Life Sciences on order-made podzolic soil. The groundwater level was at a depth of 180 cm. The distance between the rows of every orchard was 4 m. In the first orchard, the distance between the trees in the row was 2.0 m. In the remaining OR2 and OR3, after two years of cultivation, it was decided to reduce the row spacing to 1.3 m due to the large space between the trees.

Between the rows of trees, a belt of grass was maintained, and a belt under the crowns of herbicide fallow trees, where Roundup 360 SL glyphosate was applied 2–3 times as weeds appeared, Monsanto Europe N.V., Belgium). Protection against diseases and pests was based on current production recommendations in commercial sour cherry orchards. The cultivar is sensitive to cherry leaf spot caused by *Blumeriella jaapi* (Rehm) Arx. and quite resistant to other fungous diseases.

Before the establishment of the experiment, a chemical analysis of the soil was performed, and soil fertilization was carried out based on mineral content [39]. As a result of the analysis soil liming was applied at a dose of 2 t ha$^{-1}$, then soil deepening and plowing

were carried out to mix the fertilizer with the soil. In April, mustard was sown for green manure, which in June was crushed and plowed. Mustard for the same purpose was grown twice more in June and August. In November, mustard was mowed and mixed with the top layer of soil. Then, 40 t ha$^{-1}$ of manure was scattered and plowed. Two years before planting the trees, legume plants were grown for green manure and the soil was carried out. Before planting, potassium (200 kg·ha$^{-1}$ K$_2$O), phosphorus (185 kg·ha$^{-1}$ P$_2$O$_5$), and 60 t·ha$^{-1}$ manuring were applied to each orchard. In the spring before planting trees, fibering of the field was carried out, then cultivating and immediately before planting harrowing.

The trees were kept with a leader, in the form of a spindle crown.

### 2.2. Fertilization

### 2.2.1. Nitrogen Fertilization

The experiment was designed in a random block system in 4 repetitions. There were 5 trees in each repetition. One combination consisted of 20 trees. Between each repetition, there were 2 insulation trees.

The following nitrogen fertilization was used: control—without fertilization (0N), 60 kg N ha$^{-1}$ (N60), 120 kg N ha$^{-1}$ (N120).

Every year in spring between 10 and 20 April, before the trees bloomed, fertilization with ammonium nitrate (34% N) was carried out.

The course of climatic conditions during nitrogen fertilization is presented in Table 1. The average air temperature in the week prior to fertilization application exceeded 6.8 °C. During the next seven years after the procedure, the average temperatures were above 8.5 °C. It is assumed that the physiological activity of the roots is limited by the soil temperature and that only a temperature above 10 °C allows the roots to gain full activity. Only in 2010, in the week after the procedure, the temperature was below this value. In addition to temperatures, the course of precipitation in the period before and after the application of nitrogen fertilizers is important. Excessive rainfall promotes the leaching of nutrients from the soil outside the root zone of plants. However, rainfall totals were too low to cause nitrogen leaching. Overintense rainfall contributes to the physical flushing of fertilizer granules from the surface of the field, which, of course, reduces the efficiency of fertilization. On the other hand, in dry years, when the amount of rainfall is negligible and temperatures are relatively high, it is difficult to dissolve the fertilizer and transport the released components to the root zone. The highest rainfall was in 2009 and 2010 when, within 7 days after fertilization, there was precipitation of 15.4 and 10.8 mm, respectively.

**Table 1.** Precipitation and air temperature before and after nitrogen fertilization in 2008–2013.

| | | 2008 | 2009 | 2010 | 2011 | 2012 | 2013 |
|---|---|---|---|---|---|---|---|
| | | Temperature before 7 days N fertilization | | | | | |
| | Air | 6.8 | 10.7 | 9.1 | 8.9 | 7.0 | 12.9 |
| | Soil | 7.7 | 10.5 | 9.7 | 10.1 | 7.9 | 11.0 |
| | | Temperature after 7 days N fertilization | | | | | |
| | Air | 9.8 | 9.4 | 8.5 | 13.4 | 8.7 | 11.8 |
| | Soil | 12.0 | 12.5 | 9.0 | 14.2 | 10.1 | 12.8 |
| | | Precipitation before 7 days N fertilization (mm) | | | | | |
| | Sum | 31.8 | 4.0 | 0.6 | 1.6 | 8.2 | 4.2 |
| | | Precipitation after 7 days N fertilization (mm) | | | | | |
| | Sum | 0.0 | 15.4 | 10.8 | 0.0 | 1.0 | 0.8 |
| | | April | | | | | |
| Mean temperature (°C) | | 7.9 | 11.2 | 8.8 | 11.5 | 8.9 | 7.9 |
| Sum of precipitation (mm) | | 56.2 | 19.6 | 15.0 | 9.2 | 33.2 | 7.0 |



### 2.2.2. Fertilization with Other Ingredients

From the third year after planting, fertilization with other components was applied as necessarily determined on the basis of soil analysis. Every other year, magnesium lime was applied at a dose of 750 kg ha$^{-1}$ (40% CaO and 10% MgO). Every other year in autumn, 30 kg ha$^{-1}$ in the form of potassium chloride was also applied. Phosphoric fertilization was not used during the experimental period.

### 2.3. Measurements, Observations, and Analyses

### 2.3.1. Vegetative Growth

In autumn, after the end of vegetation between 10 and 20 November, the volume of the trunk was measured at the height of 30 cm from the ground. Based on volume, the cross-sectional area of the trunks (TCSA) was calculated. The difference between the measurements in subsequent years was used to calculate the TCSA increment between the years of research.

The measurement of the leaf area was carried out after the fruit harvest. A dozen or so undamaged leaves were taken randomly from each tree. A random number of tens were taken for each tree, so the measurement without a single repetition was 50 leaves. The leaves were scanned, and the image obtained was analyzed using the Digshape 1.9 program (ver.1.9.19, Cortex Nowa, Bydgoszcz, Poland), which calculated the leaf area. The result is given in cm$^2$ per leaf.

### 2.3.2. Soil Sampling

### Determination of Minerals

Each year, at the turn of July and August, soil samples were taken. The determination of the content of the P, K, Mg, and soil reaction was carried out in the laboratory of the Department of Pomology in Poznań. In each replication, four samples were taken from the herbicide belt from two layers of soil: arable 0–20 cm and sub-arable 21–40 cm, which were then poured into a container. A soil drill was used for collection. A final soil sample of 500 mL was taken from the container after mixing the individual sub-samples. From each fertilizer combination in the orchard, four soil samples with a volume of 500 mL were taken. The content of the minerals in the soil was determined according to the methods of Egner–Riehm (P and K) and Schachtschabel (Mg) [40].

The content of Cu, Mn, Zn, and Fe in the soil was set based on the standard in force in Poland.

### Determination of Nitrogen in Soil

The nitrogen determination was taken on four dates: after flowering trees in May (T1), during the intensive fruit growth at the end of June, the beginning of July (T2), after fruit harvest in August (T3), and after the end of vegetation at October (T4). The sampling method was analogous to the determination of other minerals. The samples were transported to the Department of Pomology immediately after the collection and frozen. In the frozen state, they were submitted for analysis the next day. The assessment of the content of nitrogen forms: N-NH$_4$ and N-NO$_3$ in the soil ware carried out based on the standards in force in Poland.

### 2.3.3. Leaves Sampling

Leaves for analysis were collected at the turn of July and August after fruit harvest. Samples were taken from each repetition separately. The sample consisted of 150–200 leaves. The leaves were collected from half of the annual increments at a height of 130–180 cm. Immediately after harvest, the leaves were dried in a dryer at 65–70 °C. The K and Mg content was measured with an atomic absorption spectrometer, calcium, by atomic absorption with a lanthanum concentration of 1%. The P content was measured calorimetrically in a molybdenum-vanadium mixture. The total nitrogen level was measured using the Kjeldhal

method. The mineral content in the soil and leaves was compared with the standard values applicable in Poland [41].

### 2.3.4. Determination of Pigments in Leaves

Fragments of a leaf blade weighing 0.5 g were cut from the leaves with a corkscrew, and 5 mL of DMSO (dimethyl sulfoxide) was poured. The samples were left for about 1 h in the dark, at room temperature, and then incubated at 65 °C (water bath) for 20 min. In the obtained extract after cooling, the content of chlorophyll a and b was determined spectrophotometrically.

The chlorophyll pigment content was determined using a spectrophotometer (Spekol, Carl Zeiss Jena, Jena, Germany) at the appropriate wavelength. For chlorophyll a, the absorbance of the extract was measured at a wavelength of 663 nm, for chlorophyll b at a wavelength of 645 nm, and for carotene at a wavelength of 470 nm. The amount of chlorophyll a, chlorophyll b, and the sum of chlorophyll a + b were calculated using formulas from the paper (Arnon 1949) [42].

$$\text{Chlorophyll a} = (12.7 \cdot A663 - 2.7 \cdot A645) \cdot V \cdot (1000\ W)^{-1}$$

$$\text{Chlorophyll b} = (22.9 \cdot A645 - 4.7 \cdot A663) \cdot V \cdot (1000\ W)^{-1}$$

$$\text{Sum a + b} = (20.2 \cdot A645 + 8.02 \cdot A663) \cdot V \cdot (1000\ W)^{-1}$$

$$\text{Carotenoids} = (1000 \times A470 - 1.9\ \text{chlorophyll a} - 63.14\ \text{chlorophyll b}) \times 214^{-1}$$

where: A—absorbance at a given wavelength, V—total volume of extract (cm$^3$), W—mass of the sample (g).

### 2.4. Analysis of Weather Conditions

Weather data were collected in court using an automatic weather station, iMETOS (Pessl Instruments, Weiz, Austria). The soil and air temperature at a height of 2 m above ground level and the amount of precipitation were measured. Measurements of insolation during the study period were carried out using the meteorological station in PULS (HOBO®—Plus4, made by ONSET Computers, Bourne, MA, USA). Based on the measurements, average temperatures and rainfall totals are shown (Table 2).

**Table 2.** Meteorological conditions in 2008–2013.

| Month | Precipitation (mm) | | | Temperature (°C) | | |
|---|---|---|---|---|---|---|
| | Mean 1982–2007 | Mean 2008–2013 | Change from Mean 1982–2007 in % | Mean 1982–2007 | Mean 2008–2013 | Change from Mean 1982–2007 |
| January | 30.0 | 38.0 | +26 | −0.7 | −1.6 | −0.9 |
| February | 27.7 | 16.3 | −41 | −0.1 | −1.0 | −0.9 |
| March | 34.5 | 29.7 | −14 | 3.5 | 3.0 | −0.5 |
| April | 29.0 | 33.2 | +15 | 9.1 | 10.0 | 0.8 |
| May | 45.9 | 65.6 | +43 | 14.7 | 14.0 | −0.7 |
| June | 61.4 | 69.7 | +14 | 17.2 | 17.3 | 0.1 |
| July | 72.5 | 99.1 | +37 | 19.5 | 19.5 | 0.0 |
| August | 60.5 | 63.8 | +5.0 | 18.9 | 18.8 | −0.1 |
| September | 41.1 | 47.5 | +16 | 14.1 | 13.9 | −0.2 |
| October | 30.7 | 41.1 | +34 | 9.2 | 8.3 | −0.9 |
| November | 36.0 | 39.1 | +9.0 | 3.5 | 4.7 | 1.3 |
| December | 39.9 | 39.3 | −2.0 | 0.5 | −0.4 | −0.9 |
| Total | 509.2 | 609.20 | +14 | 9.1 | 8.9 | −0.2 |

### 2.5. Statistical Analysis of Results

The results obtained were subjected to statistical analysis in Statistica 13.3 software (TIBCO Software Inc., Palo Alto, CA, USA). The means were compared using the Duncan test for a $p \leq 0.05\%$. The relationships between the features were calculated using Pearson's correlation coefficient. The principal component analysis (PCA) was used to determine the relationship between climatic conditions on nitrate nitrogen content.

## 3. Results and Discussion

### 3.1. Ammonia and Nitrate Nitrogen Content in Soil

Nitrogen fertilization had a significant impact on the nitrogen content of the soil (Figures 1 and 2). With the increase in the dose of nitrogen fertilizer, the content of ammonium and nitrate nitrogen increased. However, the significance of the differences depended on the sampling depth and the nitrogen compound. In the upper layer, no significant differences were found between the control combination and fertilization of 60 kg N ha$^{-1}$. However, in the 0–20 cm layer, nitrate nitrogen was significantly higher after applying 60 kg N ha$^{-1}$. The lack of a significant increase in nitrogen content in the soil can be explained by the fact that the trees consumed most of the nitrogen used. A similar relationship was found in the apple orchard, where the dose of 50 kg N ha$^{-1}$ had no effect on the growth of assimilable forms of nitrogen in the soil. The lack of differences was explained by the high content of organic matter in the analyzed soil of the orchard [43]. In our experiment, the value of organic material was much lower than in the cited experiments. Despite this, the dose of 60 kg N ha$^{-1}$ did not significantly increase the nitrogen content. Increasing the dose to 120 kg N ha$^{-1}$ significantly increased the nitrogen content in both soil layers. This effect of the increasing content of available forms of nitrogen indicates that the trees did not take up more fertilizer, probably caused by exceeding the fertilizing needs of trees. Nitrogen fertilization in the amount of 120 kg N ha$^{-1}$ on the entire soil surface carries a real risk of groundwater contamination. Obtained results are also confirmed by studies in the apple orchard, where a sharp increase in the overall nitrogen content in the soil occurred at a dose of 100 kg N ha$^{-1}$ in herbicide fallow belts [43]. On the other hand, there was no significant increase in nitrogen content in the turf belt because the turf stabilizes the nitrogen content in the soil [43,44]. It should be emphasized that an additional source of nitrogen is part of the mowed turf, which goes to the tree rows and undergoes mineralization, providing an additional 10–25 kg N ha$^{-1}$ annually [45], depending on the number of swaths and the temperature course. This is a significant source because, according to Kowalczyk et al., 2022 [43], which studied using different nitrogen doses, this amount can satisfy 50% of the nitrogen needed by fruit trees [45].

As a result of the mineralization of organic matter, an increase in nitrogen abundance can be observed near herbicide belts near the tree line, resulting in a lack of reaction not only to fertilization but also to nitrogen leaching [43,46]. The lack of significant differences between the combination of N0 and N60 may also be related to the presence of heterotrophic bacteria, including, among others, bacteria of the genera *Azotobacter* and *Azospirillum*, which assimilate N$_2$. Despite its small amount, it is released into the soil environment and has a major impact on soil fertility [21]. This mechanism is supported by the observation that the use of high doses of mineral nitrogen fertilizers inhibits the activity of nitrogenase and reduces the number of diazotrophic bacteria [22,24]. Therefore, its potential contribution to the soil nitrogen supply is smaller in fertilizer combinations. According to research conducted in an intensive apple orchard, the requirements of apple trees in the initial period of growth are only 8.8 to 44 kg N ha$^{-1}$ [47]. According to the integrated production methodology for apple and cherry orchards, the excessive dose of nitrogen fertilization is 100 kg N ha$^{-1}$ [8,48]. According to the results obtained and reports in the literature, nitrogen fertilization should take into account the abundance of soil organic matter, the way the soil is maintained in the orchard, and the climatic conditions during the growing season. The intensity of mineralization processes depends on temperature and humidity [22]. Temperature affects the population dynamics of soil microorganisms, biomass activity, and

the rate of mineralization of organic matter in the soil, as a result of which the content of assimilable forms of nitrogen in the soil increases [43,49]. Increase the temperature increase from 15 to 20 °C and increase the mineralization of organic matter by 38% [50]. The second important factor is soil moisture. The interaction between soil temperature and moisture in the process was demonstrated [51]. The most important role in the root system of plants is played by associative diazotrophic bacteria binding $N_2$ [24]. These include $N_2$-binding *Azospirillum* bacteria, whose activity decreases with increasing soil moisture and deterioration of soil aerobic conditions [52].

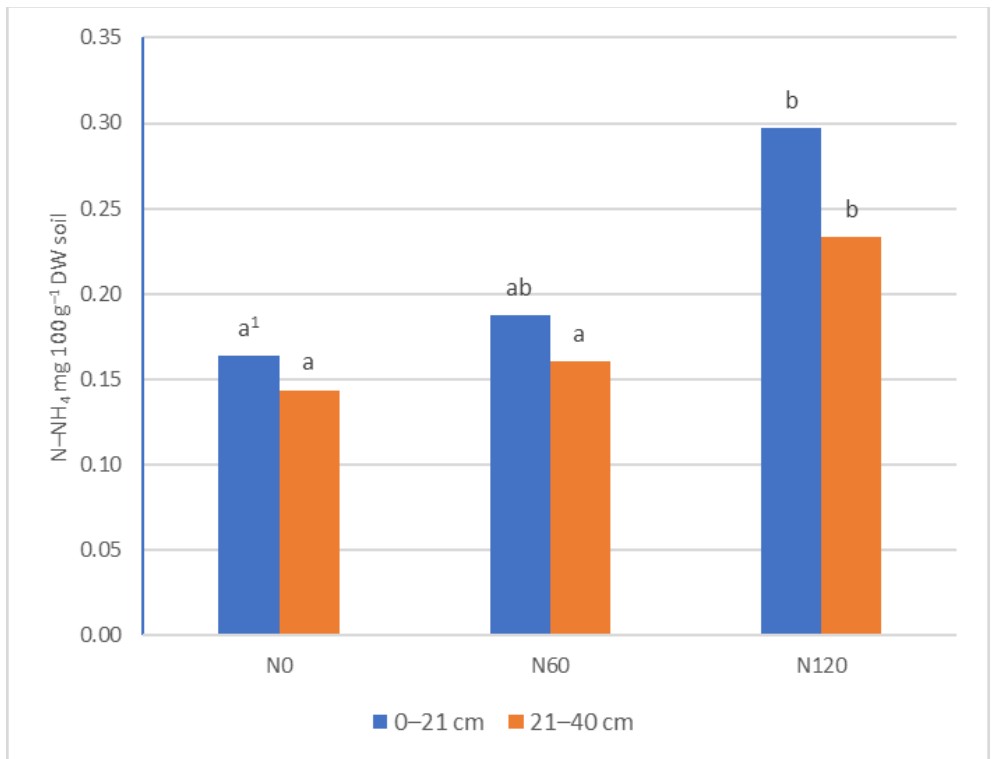

**Figure 1.** The content of N-NH$_4$ depending on nitrogen fertilization in the soil layer 0–20 cm and-21–40 cm. Statistical analysis performed separately for each soil level. [1] means that the same letters for each soil level are not significantly different at α = 0.05 (Duncan's test).

Within herbicide belts, the only factor that can stabilize the nitrogen content is the activity of soil microorganisms involved in the processes of denitrification and nitrogen uptake by the roots of trees and plants from the turf belt, which causes the variability of nitrogen content in the soil under herbicide belts is much higher than in the soil under grassland [46]. Then the nitrogen content in the grass is the result of binding by soil microorganisms in the process of decomposition of organic matter and uptake by the turf of this component for grass growth. In addition, 40% of the mowed grass is transferred to tree belts, providing nitrogen-rich organic matter. As a result, the C:N ratio is higher in the turf belt than in the herbicide fallow belt [45,53,54]. Significant differences between fertilization levels were evident in the deeper soil layer, suggesting a movement of nitrogen compounds into the soil profile under herbicide fallow, and an increase in fertilization rates leads to greater leaching of nitrates into deeper soil layers and groundwater pollution [43,55].

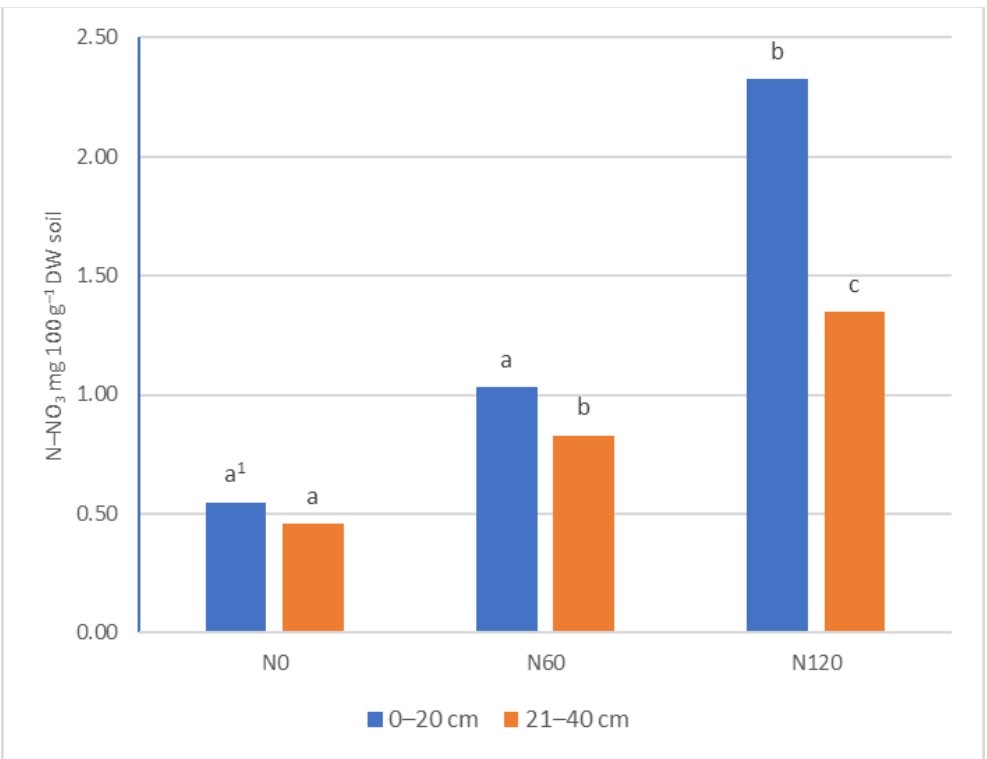

**Figure 2.** The content of N-NO₃ depending on nitrogen fertilization in the soil layer 0–20 cm and-21–40 cm. Statistical analysis performed separately for each soil level. [1] means that the same letters for each soil level are not significantly different at α = 0.05 (Duncan's test).

At the first sampling date (T1), the nitrogen content was higher than in the next three (Figures 3, 4, S1 and S2). Only in the sub-arable layer were nitrate content fluctuations observed, depending on the date of the analysis (Figure 4). The high level of nitrogen in the soil in the first term was the result of applied fertilization and the limited capacity of trees to take nitrogen in early spring because, in the early stages of development, they use nitrogen accumulated in tissues in the previous season [3]. The rate of nitrogen uptake is influenced by the temperature and development phase of trees. At 8–12 °C there is a visible increase in nitrogen absorption by the roots. Only after three weeks from the beginning of bud cracking, do apple trees take nitrogen [56]. Changes in nitrogen content also depended on the course of weather conditions. The influence of precipitation, evaporation, water balance, and soil and air temperature on ammonium and nitrogen content was evaluated. The relationship between climatic conditions and ammoniacal nitrogen content was low, sometimes seemed random, and had no effect on soil content. On the other hand, the content of nitrate was correlated with the water balance and evapotranspiration (Table 3, Figure S3). The greatest dependencies occurred in the first sampling dates. The positive water balance during sampling reduced the nitrate nitrogen content. The obtained results confirm the results of studies in which it was found that nitrate nitrogen was easily washed into deeper layers of the soil because it is not subject to biological and exchange sorption [31]. Excess rainfall in early spring, when nitrogen uptake by plants is limited and there is an excess of nitrogen in the soil profile, increases the risk of mineral nitrogen leaching from the soil [57]. The rate of mineralization of soil organic matter, depending on soil temperature, moisture, and microbial activity, will increase with increasing temperature [58,59]. An increase in mineral N leaching in cereal crops was observed with increasing temperature, in particular for sandy-loamy soils with an increase in air temperature above 2 °C [60]. Therefore, an increase in average temperature increases the availability of mineral nitrogen in the soil and contributes to its leaching [61]. The increase in evapotranspiration and desiccation of the

upper layers of the soil allows the transport of ammonium N back and reduces leaching, thus making it available to the root system through tree roots [62].

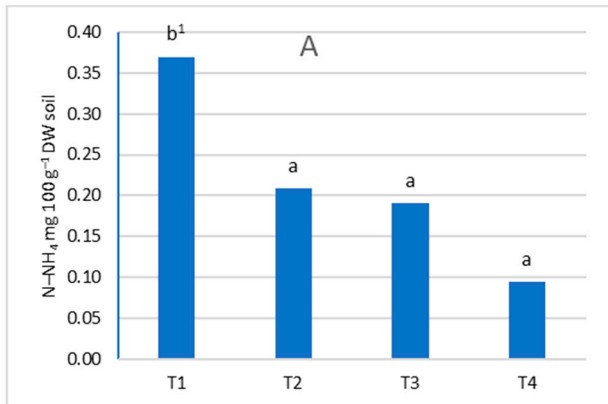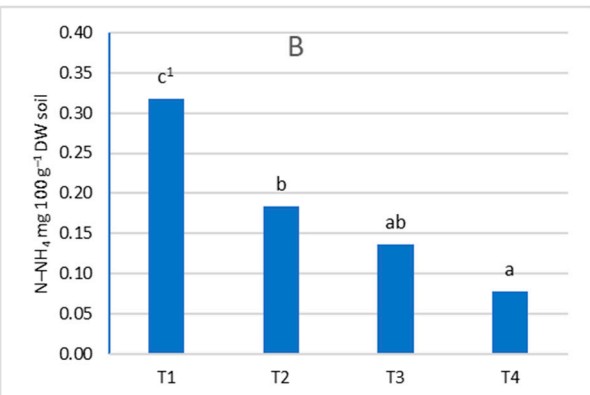

**Figure 3.** The content of N-NH$_4$ depends on the date of sampling (**A**)—in the soil layer 0–20 cm, (**B**)—in the soil layer 21–40 cm. The nitrogen determination: T1—term after flowering trees, T2—term during the intensive fruit growth, T3—term after fruit harvest in August, T4—term after the end of vegetation. [1] means that the same letters are not significantly different at $\alpha = 0.05$ (Duncan's test).

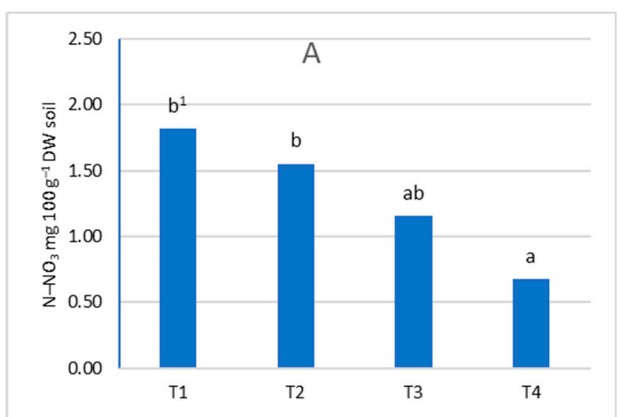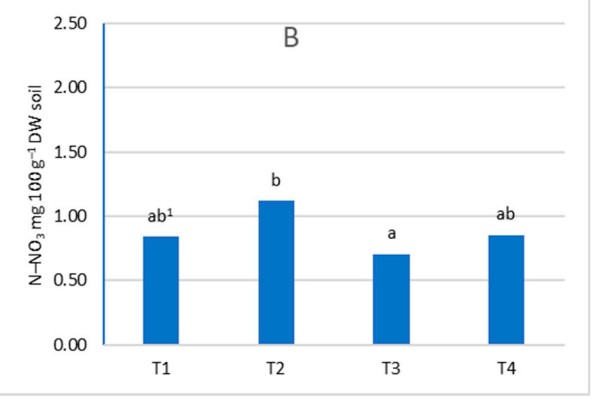

**Figure 4.** The content of N-NO$_3$ according to the date of sampling (**A**)—in the soil layer 0–20 cm, (**B**)—in the soil layer 21–40 cm. [1] means that the same letters are not significantly different at $\alpha = 0.05$ (Duncan's test).

**Table 3.** Influence of climatic conditions on ammonia nitrogen content in soil.

| | T1 | | T2 | | T3 | | T4 | |
|---|---|---|---|---|---|---|---|---|
| | **0–20 cm** | **21–40 cm** | **0–20** | **21–40** | **0–20** | **21–40** | **0–20** | **21–40** |
| P30 | 0.16 | 0.41 * | −0.51 * | −0.17 | −0.10 | 0.17 | −0.41 * | −0.45 * |
| P14 | 0.09 | 0.33 * | −0.52 * | −0.27 * | −0.11 | 0.16 | 0.24 * | 0.12 |
| Ts | 0.27 * | 0.14 | 0.49 * | 0.11 | 0.26 * | 0.26 * | 0.17 | 0.03 |
| EWT | 0.50 * | 0.51 * | −0.08 | 0.54 * | 0.25 * | 0.26 * | 0.27 * | 0.18 |
| Pds | −0.38 * | −0.32 * | −0.52 * | −0.25 * | 0.18 | 0.27 * | 0.10 | −0.06 |
| WB | −0.53 * | −0.57 * | −0.52 * | −0.33 * | −0.27 * | −0.26 * | 0.07 | −0.24 * |
| T$_{air}$ | 0.30 * | 0.20 | 0.49 * | 0.54 * | 0.01 | 0.22 | 0.06 | −0.10 |

* Significant levels $p < 0.05$. Explanations: P30—rainfall 30 days before sampling; P14—rainfall 14 days before sampling; Ts—soil temperature; EWT—evapotranspiration; Pds—precipitation on the day of harvest; WB—water balance; T$_{air}$—air temperature.

The age of the orchard had no significant effect on the nitrogen content of the soil (Figures 5, 6, S4 and S5). However, a clear trend can be observed because the highest value

of both forms of nitrogen was found in the oldest orchard (OR1). The lack of significance between orchards can be explained by the fact that a significant difference occurred in the sub-arable layer for the nitrate form, while a direction of change in content was not observed for ammoniacal nitrogen. The 0–20 cm layer was characterized by a significantly higher content of nitrate nitrogen (Figures 5B and 6B). According to the literature, fertilizing the orchard should take into account the age of the orchard, the yield, and the number of trees [8,63–65]. The higher nitrogen content in the soil of the oldest orchard may be due to decreasing nutritional needs with the age of the trees, as the trees have completed a period of intensive growth. The OR1 orchard also had the lowest tree yield [65] and the number of trees per hectare. This explains the lower nutrient requirements of trees, as with a lower density of trees per hectare of vegetation, trees compete less for nutrients and use nitrogen in a smaller percentage [66].

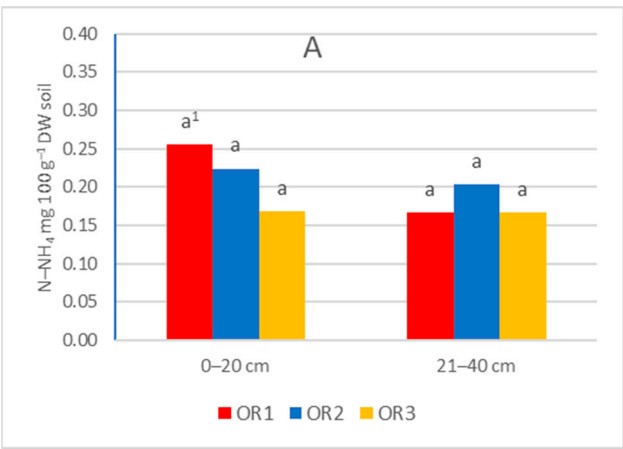 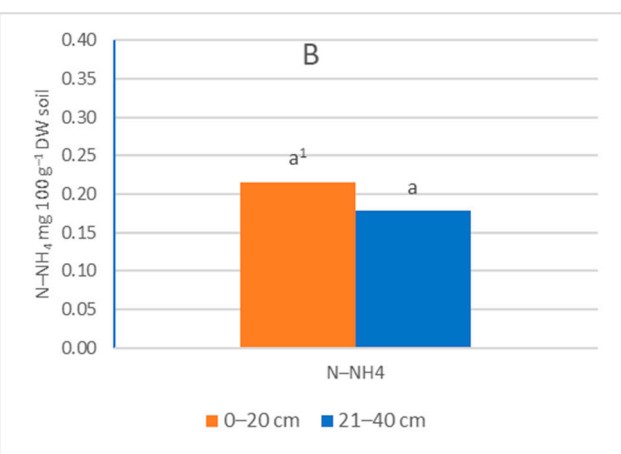

**Figure 5.** The content of N-NH$_4$ depending on the orchard (**A**)—orchard × soil layer, (**B**)—soil layer. [1] means that the same letters are not significantly different at $\alpha$ = 0.05 (Duncan's test).

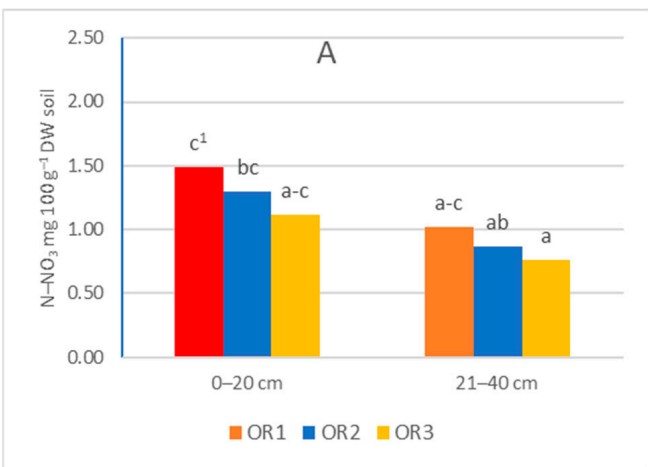 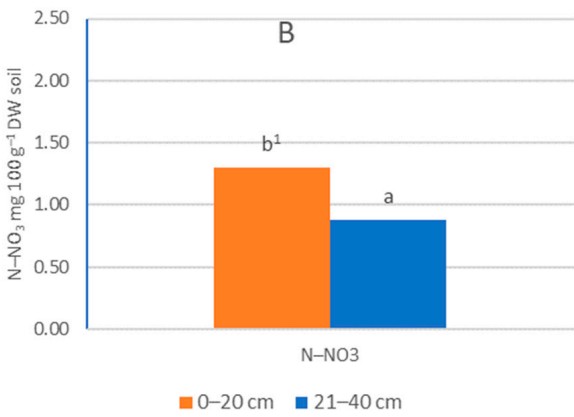

**Figure 6.** The content of N-NO$_3$ depending on the (**A**)—orchard × soil layer, (**B**)—soil layer. [1] means that the same letters are not significantly different at $\alpha$ = 0.05 (Duncan's test).

### 3.2. Content of Minerals Other Than Nitrogen in the Soil

The mineral content in the soil did not fall below the recommendations for commercial orchards. However, if differentiated nitrogen fertilization was taken into account, it was found that it also differentiated the content of other components (Tables 4 and 5).

**Table 4.** The influence of nitrogen fertilization on the content of P, K, and Mg nutrients and on pH in 2010–2013 in depth 0–20 cm.

| Orchard | Treatment | mg 100 g$^{-1}$ DW Soil | | | | | | | | pH in KCl | |
| | | P | | K | | Mg | | K/Mg | | | |
|---|---|---|---|---|---|---|---|---|---|---|---|
| OR 1 | N0 | 8.4 | b [1] | 8.5 | ab | 13.2 | b | 0.6 | bc | 6.7 | b |
| | N60 | 9.5 | c | 7.8 | a | 11.7 | a | 0.7 | c | 6.7 | b |
| | N120 | 9.6 | c | 9.8 | cd | 11.9 | a | 0.8 | de | 6.5 | ab |
| OR 2 | N0 | 7.1 | a | 7.8 | a | 11.8 | a | 0.7 | c | 6.4 | ab |
| | N60 | 8.3 | b | 8.3 | ab | 11.0 | a | 0.8 | d | 6.2 | a |
| | N120 | 8.9 | bc | 10.5 | d | 10.8 | a | 1.0 | f | 6.1 | a |
| OR 3 | N0 | 11.5 | e | 10.0 | cd | 15.5 | c | 0.9 | e | 6.6 | ab |
| | N60 | 9.6 | c | 9.2 | bc | 15.8 | c | 0.5 | a | 6.4 | ab |
| | N120 | 11.8 | e | 13.6 | e | 15.7 | c | 0.6 | ab | 6.4 | ab |
| Mean for orchard | OR 1 | 9.2 | b [2] | 8.7 | a | 12.3 | b | 0.7 | b | 6.6 | b |
| | OR 2 | 8.1 | a | 8.9 | a | 11.2 | a | 0.8 | c | 6.2 | a |
| | OR 3 | 11.0 | c | 10.9 | b | 15.6 | c | 0.7 | a | 6.4 | ab |
| Mean for treatment | N0 | 9.0 | a [3] | 8.8 | a | 13.5 | b | 0.7 | b | 6.5 | a |
| | N60 | 9.1 | a | 8.4 | a | 12.8 | a | 0.7 | a | 6.4 | a |
| | N120 | 10.1 | b | 11.3 | b | 12.8 | a | 0.8 | c | 6.3 | a |

[1] year × fertilization; the same letters are not significantly different at $\alpha = 0.05$ (Duncan's test). [2,3] the orchard and fertilization; the same letters are not significantly different at $\alpha = 0.05$ (Duncan's test).

**Table 5.** The influence of nitrogen fertilization on the content of P, K, and Mg nutrients and on pH in 2010–2013 in depth 21–40 cm.

| Orchard | Treatment | mg 100 g$^{-1}$ DW Soil | | | | | | | | pH in KCl | |
| | | P | | K | | Mg | | K/Mg | | | |
|---|---|---|---|---|---|---|---|---|---|---|---|
| OR 1 | N0 | 7.4 | cd [1] | 5.9 | a | 9.2 | bc | 0.6 | ab | 6.6 | bc |
| | N60 | 7.3 | cd | 6.8 | b | 7.8 | a | 0.8 | c–e | 6.7 | c |
| | N120 | 7.6 | cd | 7.7 | c | 7.9 | a | 0.9 | ef | 6.7 | c |
| OR 2 | N0 | 5.3 | a | 8.7 | d | 9.8 | b–d | 0.9 | ef | 6.0 | a |
| | N60 | 6.4 | b | 7.7 | c | 9.0 | b | 0.9 | d–f | 6.2 | ab |
| | N120 | 7.2 | bc | 7.6 | c | 8.0 | a | 0.9 | f | 6.3 | a–c |
| OR 3 | N0 | 8.2 | d | 8.3 | cd | 11.5 | e | 0.7 | bc | 6.2 | ab |
| | N60 | 6.3 | b | 6.5 | ab | 10.5 | d | 0.6 | a | 6.5 | bc |
| | N120 | 9.3 | e | 7.6 | c | 10.2 | cd | 0.8 | cd | 6.4 | a–c |
| Mean for orchard | OR 1 | 7.4 | b [2] | 6.8 | a | 8.3 | a | 0.8 | b | 6.7 | b |
| | OR 2 | 6.3 | a | 8.0 | c | 9.0 | b | 0.9 | c | 6.2 | a |
| | OR 3 | 7.9 | b | 7.5 | b | 10.7 | c | 0.7 | a | 6.4 | a |
| Mean for treatment | N0 | 7.0 | a [3] | 7.6 | b | 10.2 | b | 0.8 | a | 6.3 | a |
| | N60 | 6.7 | a | 7.0 | a | 9.1 | a | 0.7 | a | 6.5 | a |
| | N120 | 8.0 | b | 7.7 | b | 8.7 | a | 0.9 | b | 6.5 | a |

[1] year × fertilization; the same letters are not significantly different at $\alpha = 0.05$ (Duncan's test). [2,3] the orchard and fertilization; the same letters are not significantly different at $\alpha = 0.05$ (Duncan's test).

The P content increased for 120 kg N ha$^{-1}$ compared to the control and 60 kg N ha$^{-1}$. A similar relationship was observed in the sub-arable layer, where the phosphorus content was lower, but the nature of the changes was similar. High nitrogen fertilization also contributed to a higher potassium content in the upper soil layer. In the sub-arable layer, such a tendency was not found, ratio and at fertilization of 60 kg N ha$^{-1}$, the potassium level was significantly lower. Similar results have been obtained in many years of experience, where nitrogen fertilization increased the nitrogen content in the soil layer by 0–30 cm [67]. The magnesium content was highest in the control and nitrogen fertilization resulted in a decrease in content. The lower Mg content under the influence of fertilization may be the result of a decrease in soil pH and nitrification. Magnesium is more easily washed out

because it is less absorbed by soil colloids, moreover, $Mg^{2+}$ is not specifically bound to clay minerals [68].

The potassium-to-magnesium ratio increased after nitrogen fertilization. The trend of changes was clearly visible in the 21–40 cm layer for the dose of 120 kg N ha$^{-1}$, where the decrease in magnesium content in the 21–40 cm layer was the largest (Tables 4 and 5).

Despite the lack of significant differences in soil reaction, pH reduction was visible in the arable layer and sub-arable layer of soil. Nitrate uptake increases the pH of the external solution. In the case of ammonium form, H+ is pumped out of the cell and remains mainly outside, and therefore the pH is reduced [69,70]. This phenomenon is particularly evident in the case of nitrates since they are absorbed in large quantities compared to other types of anions [68].

### 3.3. Mineral Content in Leaves

The nitrogen content in the leaves depended on the height of nitrogen fertilization. An increase in the dose of nitrogen fertilizer resulted in an increase in the nitrogen content in the leaves (Table 6). How in all the examined orchards, a dose of 60 kg N ha$^{-1}$ was observed and 120 kg N ha$^{-1}$ significantly increased the nitrogen content in the leaves. The highest content was found at the highest nitrogen fertilization ever there were exceptions, as Vang-Petersen [71] found that increasing the dose fertilization to 162 kg N ha$^{-1}$ did not affect the increase in the nitrogen content of the leaves. According to other authors, increasing the dose of nitrogen causes an increase in the level of nitrogen in the leaves only to a certain level, above which fertilization is no longer accompanied by an increase in the content of this component in the leaves [43,46,72]. However, this regularity was not confirmed in our research because the lowest content was found in the leaves collected from the youngest orchard, where the content of assimilable forms of nitrogen in the soil was the lowest. A significant relationship was found between the overall nitrogen content in the soil and in the leaves. First of all, the ozone content in the sub-arable layer had the greatest impact on the degree of nitrogen nutrition of trees (Table 7). It was found that in addition to fertilization, several other factors have a great influence on the mineral content of the leaves. The main ones include the rootstock used [73], precipitation, and temperature [74,75]. An increase in the nitrogen content in the leaves was also found under the influence of tree cutting, whereas the intensity of tree cutting increased, the nitrogen content in the leaves increased [76].

The phosphorus content in the leaves was inversely proportional to the dose of ammonium nitrate. The highest content was in the control combination and the lowest was at a dose of 120 kg N ha$^{-1}$. A similar effect was on potassium content, where the application of nitrogen fertilization resulted in a lower content in the leaves. It should be noted, however, that the lowest content was at 60 kg N ha$^{-1}$. Similar results were obtained in earlier years, where nitrogen fertilization reduced the content of phosphorus and potassium in leaves [39]. The uptake of potassium was significantly influenced by climatic conditions. The increase in average temperature and rainfall caused an increase in the K content in the leaves (Table 7).

The magnesium content was highest with fertilization of 60 kg N ha$^{-1}$, and doubling the dose did not affect the content of this nutrient in the leaves. A similar relationship was observed with respect to calcium, where 60 kg N ha$^{-1}$ caused an increase in leaf content and a further increase in nitrogen fertilization reduced its content compared to the control treatment. The effect of nitrogen fertilization on the increase in magnesium content has been found in other experiments [74,77], but the difference between control trees and nitrogen-fertilized trees was not always significant. Climatic conditions had a significant impact on tree nutrition of trees in Mg and Ca, but the increase in temperature and rainfall during the growing season might reduce the content of Mg and Ca in the leaves (Table 7). In conditions of strong vegetative growth, a decrease in the content of components in the leaves is often observed, referred to as dilution, which distorts the results. Changes in the content of components in the leaves also depend on the intensity of yield [78]. This applies

most to N, K, Mg, Ca, Fe, Mn, and to some extent also P, where apple trees yielding more abundantly had higher contents [79]. The pad can have a significant impact on the content of ingredients. The use of strongly dwarf rootstock P22 for apple trees with intensive tree production caused nitrogen deficiency in the leaves [80]. In contrast, plums that produced intensively took only much more K, which was accumulated in the fruit while reducing the content in the leaves and perennial parts of the tree, and the level of the remaining minerals did not differ between trees that produced intensively and had no fruit [81].

**Table 6.** Effect of nitrogen fertilization on component content in leaves, depending on tree age (mean values from 2010–2013).

| Orchard | Treatment | Nutrient Content (% DW) | | | | | | | | | |
|---|---|---|---|---|---|---|---|---|---|---|---|
| | | N | | P | | K | | Ca | | Mg | |
| | N0 | 2.14 | b [1] | 0.31 | e | 1.63 | d | 2.73 | f | 0.45 | ab |
| OR 1 | N60 | 2.36 | d | 0.18 | ab | 1.24 | bc | 2.71 | f | 0.48 | bc |
| | N120 | 2.65 | f | 0.17 | a | 1.43 | c | 2.39 | de | 0.43 | a |
| | N0 | 1.96 | a | 0.28 | d | 1.08 | ab | 2.01 | a | 0.52 | de |
| OR 2 | N60 | 2.18 | bc | 0.22 | c | 1.06 | ab | 2.22 | b–d | 0.55 | ef |
| | N120 | 2.30 | cd | 0.19 | ab | 1.02 | ab | 2.13 | ab | 0.50 | cd |
| | N0 | 1.99 | a | 0.28 | d | 1.10 | ab | 2.17 | a–c | 0.55 | ef |
| OR 3 | N60 | 2.18 | bc | 0.20 | b | 0.89 | a | 2.44 | e | 0.58 | f |
| | N120 | 2.51 | e | 0.17 | ab | 0.95 | a | 2.34 | c–e | 0.52 | de |
| | OR 1 | 2.39 | b [2] | 0.22 | a | 1.43 | b | 2.61 | c | 0.45 | a |
| Mean for orchard | OR 2 | 2.14 | a | 0.23 | a | 1.06 | a | 2.12 | a | 0.52 | b |
| | OR 3 | 2.23 | a | 0.22 | a | 0.98 | a | 2.32 | b | 0.55 | c |
| | N0 | 2.03 | a [3] | 0.29 | c | 1.27 | b | 2.30 | a | 0.51 | a |
| Meant for treatment | N60 | 2.24 | b | 0.20 | b | 1.06 | a | 2.46 | b | 0.53 | b |
| | N120 | 2.49 | c | 0.17 | a | 1.13 | a | 2.29 | a | 0.48 | a |

[1] year × fertilization; the same letters are not significantly different at α = 0.05 (Duncan's test). [2,3] the orchard and fertilization; the same letters are not significantly different at α = 0.05 (Duncan's test).

**Table 7.** Influence of climatic conditions and soil mineral content on leaf content.

| | N | P | K | Ca | Mg |
|---|---|---|---|---|---|
| T | 0.01 | 0.09 | 0.52 * | −0.66 * | −0.61 * |
| T max. | −0.06 | 0.12 | 0.55 * | −0.68 * | −0.62 * |
| T min. | 0.18 | −0.17 | −0.45 * | 0.55 * | 0.50 * |
| Precipitation | 0.01 | 0.08 | 0.52 * | −0.65 * | −0.61 * |
| ET | 0.09 | −0.14 | −0.54 * | 0.67 * | 0.62 * |
| BW | 0.21 * | −0.18 | −0.39 * | 0.46 * | 0.42 * |
| N-NO$_3$ (0–20 cm) | 0.53 * | −0.08 | −0.05 | −0.07 | −0.16 |
| Total N (0–20 cm) | 0.53 * | −0.02 | −0.01 | −0.09 | −0.18 |
| N-NO$_3$ (20–40 cm) | 0.58 * | −0.19 * | 0.00 | −0.04 | −0.20 * |
| Total N (20–40 cm) | 0.60 * | −0.20 * | −0.03 | −0.04 | −0.19 |
| P (0–20 cm) | 0.31 * | −0.02 | 0.23 * | −0.19 | −0.06 |
| K (0–20 cm) | 0.40 * | −0.17 | −0.09 | 0.02 | 0.21 * |
| Mg (0–20 cm) | −0.05 | 0.08 | 0.10 | 0.04 | 0.15 |
| K/Mg (0–20 cm) | 0.28 * | −0.09 | −0.13 | −0.09 | 0.09 |
| P (20–40 cm) | 0.43 * | −0.01 | 0.31 * | −0.17 | −0.37 * |
| K (20–40 cm) | 0.24 * | −0.10 | −0.18 | −0.02 | 0.15 |
| Mg (20–40 cm) | −0.22 * | 0.21 * | −0.07 | −0.01 | 0.39 * |
| K/Mg (20–40 cm) | 0.38 * | −0.22 * | −0.05 | −0.02 | −0.19 |

* Significant levels $p < 0.05$. T—average temperature of vegetation period, Tmax.—average maximum temperature of vegetation period, Tmin.—average minimum temperature of vegetation period, ET—evapotranspiration, BW—water balance.

### 3.4. Effect of Nitrogen Fertilization on the Growth of Cherry Trees

Increasing nitrogen fertilization to 120 kg N ha$^{-1}$ resulted in greater tree growth measured by the cross-sectional area of the trunk and leaf area (Table 8). The cross-sectional area of the trunk after six years of study was the smallest in unfertilized trees, while the doses 60 kg N ha$^{-1}$ and 120 kg N ha$^{-1}$ did not cause differences. However, an interesting result was observed when comparing the increase in the cross-sectional area. The difference between 60 and 120 kg N ha$^{-1}$ of fertilization was significant taking into account all orchards combined because this variability in individual orchards was less pronounced. This is an important argument to not increase nitrogen fertilization excessively. Similar conclusions were previously reached by Fallahi and Mohan (2000) [74], who observed variation in tree growth, it was only between small doses, while excessive fertilization no longer differentiated growth [65]. This is especially noticeable in the orchard on fertile soil in the first years after planting. In general, fertilization has no effect on tree growth, and only when the root system expands into turf strips and there is a nitrogen deficit, this effect is significant [82]. In addition, the growing root system increases the growth and number of microorganisms by secreting secretions from the roots and providing food to microorganisms, accelerating the mineralization process [83]. Soil microorganisms have a C:N ratio of about 8. When the C:N ratio is between 1 and 15 [84], rapid mineralization and release of N, which is available for uptake by plants [85]. Research shows that after only 8 years of cultivation in a cherry orchard, 50% of the small roots responsible for the uptake are located in the turf belt [56]. In our experiment, however, the reaction was found earlier, which was certainly due to the low level of organic matter.

**Table 8.** The influence of nitrogen fertilization on cherry growth in 2007–2013.

| Orchard | Treatment | Leaf Area (cm$^2$) | | TCSA | | | | | |
| | | | 2007 | | 2013 | | Increase 2007–2013 | |
|---|---|---|---|---|---|---|---|---|
| | N0 | 29.0 | a [1] | 56.5 | b | 84.1 | bc | 27.7 | ab |
| OR 1 | N60 | 34.1 | bc | 68.0 | c | 102.4 | d | 34.4 | a–d |
| | N120 | 34.7 | bc | 54.9 | b | 92.1 | c | 37.2 | b–d |
| | N0 | 28.0 | a | 43.5 | a | 69.3 | a | 25.9 | a |
| OR 2 | N60 | 32.8 | b | 46.8 | a | 78.0 | ab | 31.2 | a–c |
| | N120 | 34.9 | bc | 46.7 | a | 88.6 | c | 41.9 | d |
| | N0 | 28.7 | a | 41.5 | a | 72.2 | a | 30.8 | a–c |
| OR 3 | N60 | 32.6 | b | 43.1 | a | 78.7 | ab | 35.6 | a–d |
| | N120 | 36.3 | c | 39.7 | a | 77.9 | ab | 38.2 | cd |
| | OR 1 | 32.6 | a [2] | 59.8 | c | 92.9 | b | 33.1 | a |
| Mean for orchard | OR 2 | 31.9 | a | 45.7 | b | 78.7 | a | 33.0 | a |
| | OR 3 | 32.5 | a | 41.4 | a | 76.3 | a | 34.9 | a |
| | N0 | 28.6 | a [3] | 47.1 | a | 75.2 | a | 28.1 | a |
| Mean for treatment | N60 | 33.2 | b | 52.6 | b | 86.4 | b | 33.8 | b |
| | N120 | 35.3 | c | 47.1 | a | 86.2 | b | 39.1 | c |

[1] year × fertilization; the same letters are not significantly different at $\alpha = 0.05$ (Duncan's test). [2,3] the orchard and fertilization; the same letters are not significantly different at $\alpha = 0.05$ (Duncan's test).

The leaf area was significantly higher with nitrogen fertilization and was highest with 120 kg N ha$^{-1}$. A similar relationship was found in the mass of leaves. However, the SLA (specific leaf area—cm$^2$/g) did not differ significantly between the fertilization combinations and was very similar (Figure 7). Therefore, the only explanation is that applied nitrogen fertilization increases the surface of cherry leaves and does not increase their thickness. More so because a similar relationship has already been found in research related to soil and foliar fertilization of almond trees [25]. Similar results were also obtained when conducting research on a tropical species with edible fruits *Dimocarpus longan*, where also an increased dose of nitrogen caused the growth of the leaf blade and fresh and dry

leaf mass but had no effect on SLA [26,86]. Only apple trees reacted to an increase in the specific weight of leaves under the influence of fertilization, although no differences between nitrogen doses were found. The increase in SLW (specific leaf weight—mg/cm$^2$) was attributed to a change in leaf morphology or an increase in soluble sugars or starch [87].

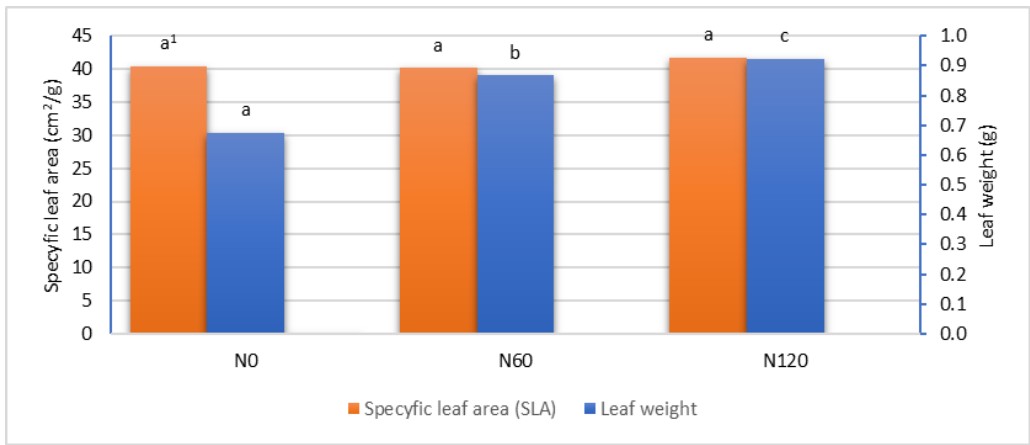

**Figure 7.** Effect of nitrogen fertilization on the leaf surface (cm$^2$) and SLW (leaf area per fresh weight) of sour cherries, average for 2012–2013. [1] means that the same letters are not significantly different at $\alpha = 0.05$ (Duncan's test).

Fruit trees are perennial plants, and the effect of fertilization applied in the current year may be visible in the following or subsequent years. The sour cherry cultivar 'Łutówka' belongs to the cherry cultivars that bear fruit on last year's shoots. Their number and length will have a significant impact on the amount and size of the crop in the following year. An important factor is also the use of appropriate tree cutting, which must stimulate the tree to produce new growths, and at the same time, the cutting intensity should not be too high to limit the number of fruiting shoots. Therefore, appropriate nitrogen fertilization allows you to maintain a balance between the growth and yielding of such fruiting trees. It is also necessary to take into account tree spacing, which in intensive cherry orchards is much smaller than in old orchards, where the spacing between rows was at least 4.5 m and in a row 3 m. Such a spacing allowed the formation of trees with a larger crown, but with such a spacing, the yield per unit area is much lower than from intensive orchards. The crown in intensive orchards has the shape of a spindle, and for mechanical harvesting, such a crown is better due to the high cost of labor, and smaller sizes are also better [88].

Increasing fertilization, apart from affecting the growth of trees, causes a change in quality parameters such as coloring. Sour cherries are primarily used for processing, where color, size, extract content, and acidity are crucial [65,89]. Trees that, as a result of poor nitrogen nutrition and improper cutting, have a low yield and at the same time become unprofitable. That is why proper nitrogen fertilization, which is responsible for the vegetative growth of trees, is so important.

*3.5. Pigment Content in Leaves*

Nitrogen fertilization did not have a significant effect on the chlorophyll pigment in the leaves ($p = 2899$). However, it should be emphasized that chlorophyll a was higher at the highest dose of nitrogen. The chlorophyll b content was higher when nitrogen fertilization was applied to 60 kg N ha$^{-1}$ and 120 kg N ha$^{-1}$. Carotenoid levels also increased under nitrogen fertilization, but the differences between the combinations were not significant. Similar results were obtained using nitrogen fertilization in apple trees, where the dose of 80 kg N ha$^{-1}$ and 250 kg N ha$^{-1}$ caused an increase in chlorophyll content, but only chlorophyll a changed during the growing season. The increase occurred until mid-July and then decreased. In contrast, chlorophyll b and carotenoids did not change during the growing season. N fertilization resulted in much higher photosynthetic pigments in

the late season; it can be assumed that the application of N contributed to delayed leaf aging. The highest values were recorded at the highest dose of nitrogen fertilization [73]. The use of nitrogen fertilization reduces the C:N ratio and accelerates the mineralization process and the release of nitrogen available to trees, which extends the growing season and delays leaf aging [84]. Nitrogen is needed to build chlorophyll, and its presence affects its production. Of the more than 130 atoms that build chlorophyll, only 4 are nitrogen atoms that form a porphyrin ring. Nitrogen deficiency is most pronounced because it inhibits the formation and development of chloroplasts. Chlorophyll synthesis is inhibited when the a/b ratio of chlorophylls is lower. The synthesis of carotenoids is also inhibited, which may correlate with chlorophyll synthesis [90]. Research conducted in apple trees confirms that the increase in nitrogen fertilization increased the amount of chlorophyll a and b. However, the difference in total chlorophyll (chlorophyll a + b) between fertilization of 50 kg N ha$^{-1}$ and 100 kg N ha$^{-1}$ was negligible [91]. A similar effect was found in another study, with the effect being more pronounced when high nitrogen levels (250 kg N ha$^{-1}$) were applied at the end of the season [72]. Increasing the chlorophyll content can be achieved by foliar use of substances containing nutrients together with 5-aminolevulinic acid (ALA), which is a key precursor in the biosynthesis of porphyrins such as chlorophyll [92].

With the age of the orchard, the level of pigments in the leaves increased (Figure 8). Long-term studies conducted on apple trees have given different results. The highest values were found in the first 6 years of intensive growth, and in subsequent periods the overall level of chlorophyll decreased [93]. Studies of the amount of pigments contained in the assimilation apparatus of Scots pine (*Pinus silvestris* L.) showed a clear increase in their content with the age of the needles and a decrease with the increasing age of trees [94]. However, this trend was in chlorophyll pigments a and b until 2012. In 2013, the content of chlorophyll a and chlorophyll b was significantly lower (Figure 8). These results can be explained by the lack of sufficient water, which causes a decrease in chlorophyll, a violation of enzyme balance, and a significant decrease in nitrate reductase activity [95]. In conditions of greater moisture of cells, the photosynthetic apparatus of the leaf is "diluted" [96]. Analyzing the course of the climatic conditions in the period prior to the analysis of chlorophyll content, a significant relationship was found between the amount of precipitation in July and the content of chlorophyll (Figure 9). Studies of tree species showed that the leaves of irrigated trees had a lower content of total chlorophyll compared to leaves taken from non-irrigated trees [97]. A similar trend of increasing content in subsequent years of research was found in carotenoids, but there was no decrease in carotenoid content in 2013 and no correlation with precipitation in the period prior to sampling. In studies conducted on cherry and plum leaves, a higher concentration of the pigment tested was noted during the harvest period, regardless of the level of irrigation and fertilization [97,98]. This phenomenon can be explained by the enormous mobility of chlorophyll pigments. The plant, wanting to preserve these pigments during fruit ripening in the period preceding harvest, withdraws pigments to leaves, while in autumn, during the period of yellowing of the leaves, the pigments are withdrawn to the permanent tree [99]. The decrease in chlorophyll content may be associated with lower yields in 2013 [65], accompanied by a lower photosynthesis intensity and chlorophyll content [90]. An additional factor that affects the chlorophyll content is tree cutting. Both the timing of the cut and the intensity affect the chlorophyll content. In the experiment carried out after the high tree yield in 2012, moderate summer pruning was applied, which could have affected pigments in the leaves. This is confirmed by studies where the pruning of peaches in summer caused a decrease in chlorophyll a and b, while pruning in winter increased the chlorophyll content [100]. In another experiment, cutting mango trees resulted in the highest chlorophyll content, while light pruning had no effect on the overall pigment content [101].

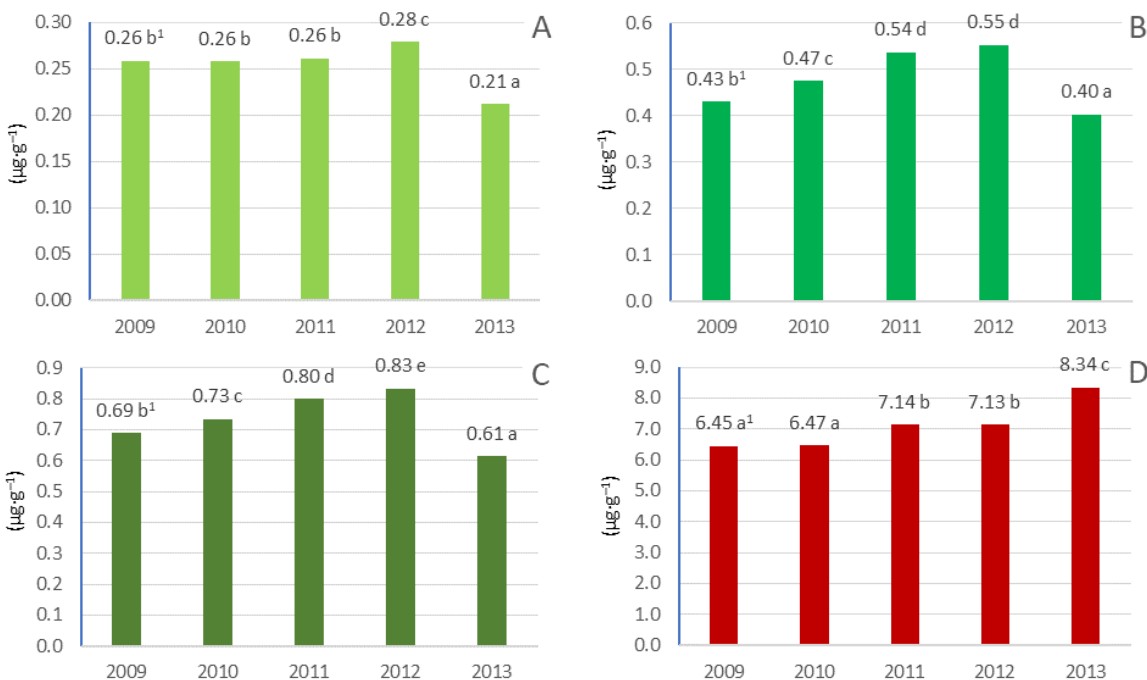

**Figure 8.** The content of chlorophyll pigments in the leaves depends on the year of research. (**A**)—chlorophyll a, (**B**)—chlorophyll b, (**C**)—chlorophyll a + b, (**D**)—carotenoids. [1] means that the same letters are not significantly different at $\alpha = 0.05$ (Duncan's test).

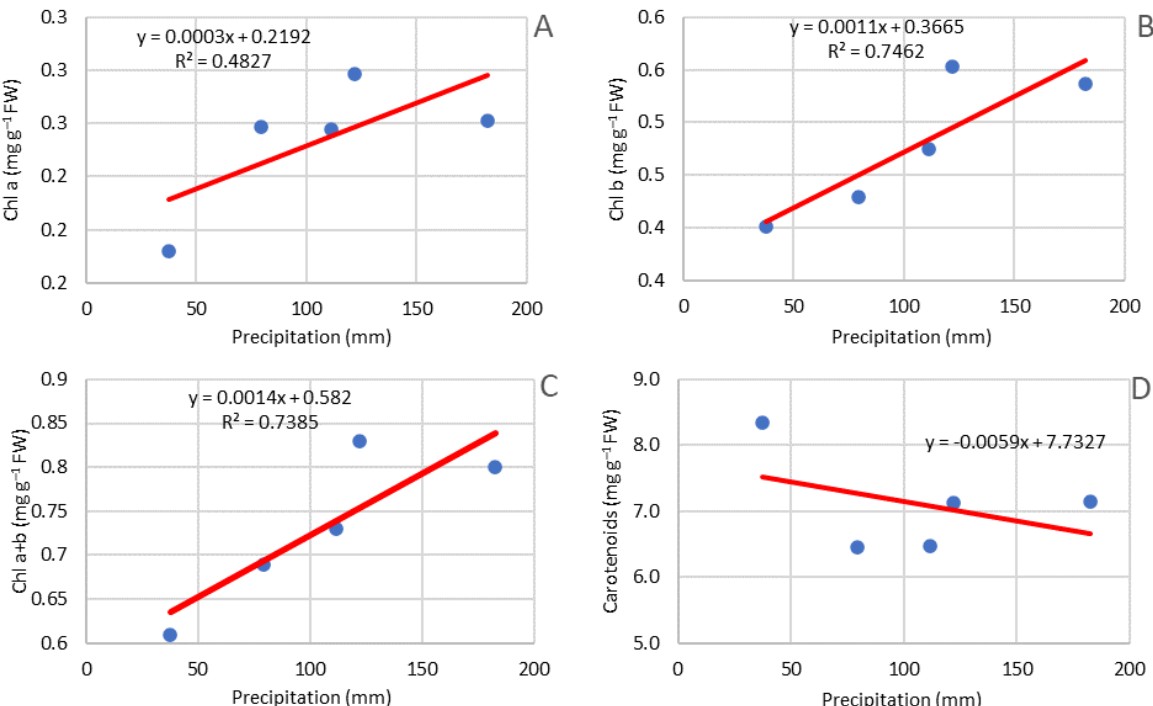

**Figure 9.** Effect of precipitation in July on the content of chlorophyll pigments. (**A**)—chlorophyll a, (**B**)—chlorophyll b, (**C**)—chlorophyll a + b, (**D**)—carotenoids.

## 4. Conclusions

Nitrogen fertilization at the beginning of the growing season impacted the content of ammoniacal and nitrate nitrogen in the soil. The nitrogen content is influenced by the course of the weather, but above all, by the uptake by trees. These changes reflect the fertilizer needs of the studied species. Nitrate nitrogen increased with the applied

dose regardless of the sampling level, but the differences between the combinations were significant only in the sub-arable layer. Probably the use of N60 covered the fertilizer needs of cherry trees. Only the increase in the dose to N120 significantly increased the nitrogen content in the soil.

The highest nitrogen content in the soil was found in the first sampling period at the beginning of vegetation when ammonium nitrate was applied. During this time, trees benefit from nitrogen that was accumulated in the tissues of the tree after being taken in the previous season. The ammonium form content decreased on subsequent sampling dates. A similar trend was observed for the nitrate form in the topsoil layer, while in the sub-arable layer, the nitrate nitrogen content varied depending on the timing of the samples. An important factor that influenced the content was the course of climatic conditions, such as precipitation and temperature.

The age of the orchard had no significant effect on the nitrogen content in the soil.

Regardless of the nitrogen form, the content was higher in the arable layer of the soil, but the difference in content was significant only for the ammonium form.

High nitrogen fertilization increased the content of phosphorus and potassium and decreased the magnesium content in the topsoil layer, resulting in an increase in the K/Mg ratio. It also slightly (insignificantly) reduced the pH of the upper soil layer. Similar changes were observed in the sublayer for phosphorus and magnesium, while the K level was significantly lower with N60 fertilization with no significant differences between the control combination and the highest fertilization dose. The pH of the soil was higher with nitrogen fertilization, but the differences were not significant.

The nitrogen content in the leaves increased, whereas the content of phosphorus and potassium decreased with increasing doses of nitrogen. The highest magnesium and calcium content was after N60 fertilization; increasing the dose to N120 reduced their content, which can be explained by stronger vegetative growth and the dilution of ingredients.

Nitrogen fertilization increased the vegetative growth of trees measured by leaf area, trunk cross-sectional area, and their growth during the research period. However, SLA ($cm^2/g$) did not differ significantly between combinations, because the thickness of the leaves in the control and fertilized ammonium nitrate combinations was the same, regardless of the dose used.

The chlorophyll content did not depend on nitrogen dosage. However, it should be emphasized that the overall chlorophyll content in the leaves was higher with nitrogen fertilization. The course of the weather, especially precipitation, which caused leaf growth, had a greater impact on the chlorophyll content. Not without significance is also the intensity of yielding and the pruning of trees in this cherry cultivar, which bears fruit on annual shoots, and pruning is carried out in the summer after harvest.

**Supplementary Materials:** The following supporting information can be downloaded at: https://www.mdpi.com/article/10.3390/agriculture13030578/s1, Figure S1: The content of N-NH$_4$ depending on the term and nitrogen fertilization A—in the soil layer 0–20 cm, B- soil layer 21–40 cm. The nitrogen determination: T1—term after flowering trees, T2—term during the intensive fruit growth, T3 term after fruit harvest in August, T4 term after the end of vegetation; Figure S2: The content of N-NO$_3$ depending on the term and nitrogen fertilization A in the soil layer 0–20 cm, B in the soil layer 21-40 cm; Figure S3: Influence of climatic conditions on the nitrate nitrogen content in the soil according to the sampling time, as shown by PCA. A—sampling time T1, B—sampling time T2, C—sampling time T3, D—sampling time T4. WB—water balance, P1—sum precipitation 30 days before sampling, P2 sum precipitation 14 days before sampling, TA—average temperature 30 days before sampling, TS—soil temperature. ET—evapotranspiration; Figure S4: The content of N-NH$_4$ depending on the year of research A—in the soil layer 0–20 cm, B—in the soil layer 21–40 cm; Figure S5: The content of N-NO$_3$ according to the year of research A—in the soil layer 0–20 cm, B- in the soil layer 21–40 cm; Figure S6: The content of pigments in the leaves depends on nitrogen fertilization. A—chlorophyll a, B—chlorophyll b, C—chlorophyll a + b, D—Carotenoids.

**Author Contributions:** Conceptualization, K.R.; methodology, K.R.; software, K.R.; validation, K.R. and G.P.Ł.; investigation, K.R.; resources, K.R. and G.P.Ł.; data curation, K.R.; writing—original draft preparation, K.R. and G.P.Ł.; writing—review and editing, K.R. and G.P.Ł.; visualization, K.R.; supervision, K.R.; project administration, K.R.; funding acquisition, K.R. All authors have read and agreed to the published version of the manuscript.

**Funding:** The publication was co-financed within the framework of the "Regional Initiative Excellence" program implemented at the initiative of the Polish Ministry of Science and Higher Education in 2019–2023 (No. 005/RID/2018/19), financing amount: PLN 12,000,000.

**Institutional Review Board Statement:** Not applicable.

**Informed Consent Statement:** Not applicable.

**Data Availability Statement:** Not applicable.

**Conflicts of Interest:** The authors declare no conflict of interest.

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
