# Peer review of "Effect of Nitrogen Fertilization on Tree Growth and Nutrient Content in Soil and Cherry Leaves (Prunus cerasus L.)"

_agriculture, doi:10.3390/agriculture13030578_

Round 1
Reviewer 1 Report
Dear authors,
Thank you for the opportunity to review this Manuscript (Effect of Nitrogen Fertilization on Tree Growth and Mineral Content in Soil and Cherry Leaves (Prunus cerasus L.)). The purpose of the study was to evaluate the impact of different doses of nitrogen fertilization on the content of the basic element in the soil and the leaves and its influence on selected cherry growth parameters. The authors have a great dataset with information of cherry growth between 2008 and 2013. However, the quality of the Figures is low and the data set is old (10 years ago). There is some aspect that should be reviewed by authors.
PLEASE, THE EDITIONS SHOULD BE ADDED TO THE MANUSCRIPT AND ANSWERED IN THE LETTER
The title is good. However, the authors could substitute “Mineral Content” for “Nutrient contents”
The abstract is clear and brings all information required. I suggest one last sentence “Based on results, conclude that XXXXXXXXXXXXXXXXXXXXXX”
The introduction is good. No suggestions
The goals are clear.
In the “Materials and Methods” and “Results and Discussion” the topics are clear and help to understand and follow the study.
In the Material and Methods.
Explain the soil management in spring 1999 (OR1), 2001 (OR2), and 2002 (OR3) on the experimental farm
Give information about the fertilizer application each year. For example, was there rainfall together with that application? Explain all factors that can alter the N efficiency.
Lines 163: The treatments should be presented in a paragraph and not as topics.
Ammonium nitrate was applied on the soil surface?
Is there any kind of additional fertilization during the study
The equation should be edited according to the journal
Figures 1 and 2 have low quality.
There is no need for climate information in Tables and Figures. The authors could present just as a Figure.
Results and discussion
The authors could present the correlation between Ammonia and Nitrate Nitrogen Content in Soil
In lines 251 and 276, the paragraphs are so long. Edit it.
Figures 3 and 4 have low quality. They could present as a Figure only
The data is not explored. The data is good, but the authors did not present it properly
Figures 6 and 7, which are the treatments? The Figure must be clear and there is no information on what is “T1”, for example
The results in Figure 9 are not explored properly. In general, there is low quality, and results are not explored properly.
The conclusion is long and confuse. The authors should give a clear conclusion to the readers.
Reviewer 2 Report
Generally,
1. very informative work! The author needs to discuss the motive behind this work, as there are many relevant studies. Also, the reason for choosing Cherry trees.
2. The article size is big compared to any research article. There is lot of good information, but sometimes it is repetitive and overwhelming! The authors need to shorten it and make it more precise.
3. I missed the detail explanation regarding the role of organic matter in different cases and the C:N ratio. The author also mentioned the diseases caused by over-fertilization. Did they find any diseases in their study?
4. It is quite surprising that this article has 17 figures and 7 tables. There are many unnecessary figures and tables that just show repetitive information. I would recommend the authors to reduce some of them or move them to the supplementary document. Also, many times authors just mentioned the table or figure number only, instead of mentioning the treatment they were discussing from the figure/table.
5. Few references need to be checked again based on the provided information. And some small punctuation mistakes are there, fixing those will improve the quality of the work!

Round 2
Reviewer 1 Report
The manuscript can be published.